# Socioeconomic disparities in diabetes prevalence and management among the adult population in Bangladesh

Karar Zunaid Ahsan[1]* , Afrin Iqbal[2] , Kanta Jamil[3] , M. Moinuddin Haider[4‡] , Shusmita Hossain Khan[5‡], Nitai Chakraborty[5,6‡], Peter Kim Streatfield[4]

1 Public Health Leadership Program, Gillings School of Global Public Health, University of North Carolina at Chapel Hill, Chapel Hill, North Carolina, United States of America, 2 Maternal and Child Health Division (MCHD), International Centre for Diarrhoeal Disease Research, Bangladesh (icddr,b), Dhaka, Bangladesh, 3 IAP World Services, Arlington, Virginia, United States of America, 4 Health Systems and Population Studies Division (HSPSD), International Centre for Diarrhoeal Disease Research, Bangladesh (icddr,b), Dhaka, Bangladesh, 5 Data for Impact (D4I), Carolina Population Center, University of North Carolina at Chapel Hill, Chapel Hill, North Carolina, United States of America, 6 Department of Statistics, University of Dhaka, Dhaka, Bangladesh

☯ These authors contributed equally to this work.
‡ MMH, SHK and NC also contributed equally to this work.
* zunaid@email.unc.edu

**Data Availability Statement:** The survey data reported in this manuscript are publicly available via the DHS Program. The BDHS 2017-18 datasets are freely available from the following website:

## Abstract

### Background

Diabetes, one of the major metabolic disorders, is rising in Bangladesh. Studies indicate there is inequality in prevalence and care-seeking behavior, which requires further exploration to understand the socioeconomic disparities in the pathophysiology of diabetes. This study examined the latest nationally representative estimates of diabetes prevalence, awareness, and management among adults aged 18 years and above in Bangladesh and its association with socioeconomic status in 2017–18.

### Methods

We used the 2017–18 Bangladesh Demographic and Health Survey data. Diabetic status of 12,092 adults aged 18 years and above was measured in the survey using fasting plasma glucose levels. We applied multivariate logistic regressions to examine the role of socioeconomic status on diabetes prevalence, awareness, and management, after controlling for relevant covariates.

### Results

Overall, 10% of adults had diabetes in Bangladesh in 2017–18, with the highest prevalence of 16% in the age group 55–64 years. Our analyses found statistically significant disparities by socioeconomic status in the prevalence of diabetes as well as the person's awareness of his/her diabetic condition. However, the effect of socioeconomic status on receiving anti-diabetic medication only approached significance (p = 0.07), and we found no significant association between socioeconomic status and control of diabetes.

http://dhsprogram.com/data/available-datasets.cfm.

**Funding:** This work was supported by USAID's Research for Decision Makers (RDM) activity, cooperative agreement no. AID-388-A-17-00006 [GR: 154601546], and Data for Impact (D4I) associate award no. 7200AA18LA00008.

**Competing interests:** The authors have declared that no competing interests exist.

## Conclusions

We expect to see an 'accumulation' of the number of people with diabetes to continue in the coming years. The rising prevalence of diabetes is only the tip of an iceberg; a large number of people with uncontrolled diabetes and a lack of awareness of their condition will lead to increased morbidity and mortality, and that could be the real threat. Immediate measures to increase screening coverage and exploration of poor control of diabetes are required to mitigate the situation.

## Introduction

Diabetes Mellitus is one of the major metabolic disorders where glucose metabolism is disrupted, resulting in elevated levels of blood glucose (or blood sugar). It is defined as a fasting plasma glucose of 7.0 mmol/L (or 126 mg/dL) or above [1]. Among several types of diabetes, type 1, type 2, and gestational diabetes mellitus (GDM) are of major concern worldwide—type 1 diabetes is caused by an autoimmune reaction in the body where the immune system attacks the cells of the pancreas that produce insulin and, as a result, little or no insulin is produced in the body; type 2 diabetes occurs when the cells in our body become resistant to insulin resulting in prevention of the absorption of sugars; GDM is typically associated with high blood glucose levels occurring later in pregnancy and usually disappears after the pregnancy ends. Type 2 diabetes accounts for approximately 90% of the total diabetes burden, and the continued rise in diabetes prevalence globally is largely due to an upsurge in type 2 diabetes and related risk factors such as overweight/obesity, physical inactivity, and calorie-dense/unhealthy diet, as a result of urbanization and sedentary lifestyles [1, 2].

According to the International Diabetes Federation (IDF), there were an estimated 151 million people aged 20–79 years with type 2 diabetes in the year 2000 globally. This number has tripled to 463 million by 2019 and will continue to rise by approximately 9 million people every year until the mid-century [1]. Any type of uncontrolled and prolonged diabetes is associated with several complications, and it can damage the small and large blood vessels, heart, eyes, kidneys, and nerves over time. While some complications very specific to diabetes (e.g., retinopathy, nephropathy, neuropathy) can compromise the quality of life, other complications such as cardiovascular and cerebrovascular diseases are associated with premature deaths [3, 4]. For the reasons of increasingly early-onset and chronically long duration, type 2 diabetes accounts for 1.8% (1.02 million) of annual deaths globally, but for 2.3% of disability-adjusted life years lost (57.4 million disability-adjusted life years) in 2017 [5]. Globally total health expenditure for diabetes was estimated to be US $760 billion per year, primarily because of the chronic long-term morbidity associated with it [1].

Bangladesh is among the top ten countries in the world for the burden of diabetes, with around 8.4 million people with this chronic condition [1]. Two nationally representative surveys conducted in 2010 among adults 25 years or above and in 2018 among adults age 18–69 years reported national diabetes prevalence of 3.9% and 8.3%, respectively [6]. According to the Bangladesh Demographic and Health Survey (BDHS), diabetes prevalence among adults aged 35 years and above increased from 11.2% to 13.8% for females and 10.7% to 13.7% for males between 2011 and 2017–18 [7, 8]. Both the BDHS rounds show a slightly higher prevalence in females except for the 65+ age group. Approximately 57% of all women in Bangladesh belong to the reproductive age group of 15–49 years [9]. These women are subject to an

additional risk of gestational diabetes, which may convert to type 2 diabetes, adding to the already high proportion of females with diabetes and related complicated cases in the country [10, 11].

The probability of dying prematurely from four major non-communicable diseases (NCDs), including diabetes, between ages 30 and 70 years has declined in many countries. Still, Bangladesh remains one of the only 20 countries where mortality has stagnated or increased from 2010 to 2016 [12]. It is often observed that undiagnosed cases present with more serious complications leading to premature death than diagnosed cases who are receiving treatment [13]. While the recent IDF estimates showed an alarming 56% of people with diabetes in Bangladesh to be undiagnosed [1], some nationally representative surveys reveal a similar picture. The 2017–18 BDHS showed 59% of females and 61% of males aged 18 years or above were unaware of their diabetic status [8]. The National STEPS Survey for NCD Risk Factors in Bangladesh 2018 also showed that 51% of people with diabetes were unaware of their condition [6]. Even among the adults aged 18 years or above who were diagnosed for having diabetes and were getting treatment, 25% of females and 22% of males did not have their blood glucose level under control [8]. The level of unawareness and uncontrolled blood glucose level among people with diabetes are major national public health concern and yet, have been relatively less prioritized.

There are visible inequalities in care-seeking and continuing treatment for chronic conditions such as diabetes. Several lifestyle behaviors, as well as knowledge and accessibility of healthcare, often drive the decision to get diagnosed and these factors are related to an individual's socioeconomic status [14]. Several studies showed that those who are economically disadvantaged, or from a marginalized portion of the community tend to suffer from more grave complications related to NCDs, including diabetes, often leading to death [15]. While the role of individual-level socioeconomic characteristics in the pathophysiology of diabetes has been studied extensively in high-income countries, few studies have explored this association in low- and middle-income countries [16]. Particularly for Bangladesh, studies published in the last ten years examining the association between socioeconomic status and diabetes were either using the 2011 BDHS data [17–23] or were based on small geographic locations [24, 25].

This study aimed to examine the latest nationally representative estimates of diabetes prevalence, awareness, and management among adults aged 18 years and above in Bangladesh and its association with socioeconomic status in 2017–18.

## Methods

### Study population and sampling

For this study, data from a nationally representative survey conducted between October 2017 and March 2018 was used. The 2017–18 BDHS is the eighth national survey, which reported the demographic and health status of women, men, and children from a nationally representative sample. The survey collected data from 20,250 households using a two-stage stratified cluster sampling process from 675 clusters in the first stage. A systematic sample of 30 households per cluster was selected in the second stage. BDHS provide statistically representative estimates of key demographic and health variables for the whole country, for urban and rural separately, and for each of the eight divisions. Details of the survey methodology can be found elsewhere [8].

The 2017–18 BDHS included measurements of biomarkers for the adult population. Blood pressure and blood glucose tests were done on all adults who were18 years and above in a quarter of the sampled households. BDHS also included height and weight measurements of all adults who were eligible for measuring blood pressure and testing blood glucose. To

measure blood specimens for glucose testing, the respondents were asked to fast overnight for eight hours. They were visited early on the next day, and made sure that they had not broken their fast before the test. In most cases, the collection of blood specimens for glucose testing required two visits to a household. The HemoCue 201 RT analyzer was used to measure blood glucose level (in millimoles per liter—mmol/L). Capillary blood was obtained from the middle or ring finger of the respondent and those who gave their full consent to participate in blood glucose measurements were included. The survey excluded pregnant cases, which brought the final study sample size to 12,092 (6,913 women and 5,179 men) respondents aged 18 years and above (see S1 Fig in S1 File for the flowchart of analytic sample).

## Outcome variables

The outcome variables for the study were: prevalence of diabetes, awareness of the respondent regarding their diabetic state, treatment status, and glycemic control among the people with diabetes. Respondents were identified as having diabetes if their fasting plasma glucose value was greater than or equal to 7.0 mmol/L on the day of the survey, or they were currently taking any medication for lowering blood glucose levels. Awareness was defined as the respondents with diabetes being aware that they have diabetes, conveyed to them by a physician or any health care provider. Treatment was defined as the respondents with diabetes who reported that they were currently taking any medication for diabetes on the day of the survey. Glycemic control was defined as the respondents with diabetes who had blood glucose below the threshold of 7.0 mmol/L while taking diabetes medication on the day of the survey.

## Explanatory variables

While selecting the explanatory variables, we primarily used a conceptual framework developed by the American Diabetes Association and the availability of data in BDHS [26]. The variables included demographic factors (age and sex of the respondent), socioeconomic factors (household wealth quintile, educational status, place of residence), and biological factors (overweight/obesity, elevated blood pressure level).

Age of the respondents was self-reported, which was divided into five categories. Socioeconomic status of the respondents was calculated based on household wealth quintiles using principal component analysis of household assets and amenities—respondents belonging to the lowest two wealth quintiles were categorized as low, 3rd quintile as middle, and the highest two quintiles as high socioecoinomic status for the analysis. Level of educational attainment was divided into four different categories. Place of residence was categorized into urban or rural areas. Body mass index (BMI) was categorized into four groups—underweight if BMI is less than 18.5 kg/m$^2$, normal if BMI is between 18.5 and 24.9 kg/m$^2$, overweight or obese if BMI is 25 kg/m$^2$ or above, and unknown for the respondents whose height and/or weight data were not available. Hypertensive was a binary variable where the respondent either had hypertension (i.e., systolic blood pressure was on or above 140 mmHg and/or diastolic blood pressure was on or above 90 mmHg on the day of the survey) or not. Following the definition used in the DHS, we also considered respondents as hypertensive who had a diagnosis of hypertension and currently taking antihypertensive medication, irrespective of their blood preassure status.

Conceptually meaningful two-way interaction terms of variables in the final main effects model were assessed with the aim of keeping significant ones, and no discernible pattern was found for significant ($p < 0.05$) interaction between potential covariates. In addition, we checked for multicollinearity using both variance inflation factor (VIF) and multicollinearity

condition number and carried out stratum-specific measures of association to identify effect modification in the model.

## Statistical analyses

Univariate analyses of the explanatory variables were performed and presented in terms of frequencies. Later, bivariate analyses (cross-tabulations) between the outcome variables and individual covariates were carried out, followed by Rao-Scott second-order correction to Pearson's Chi-square test to account for the complex survey design [27, 28]. The chi-squared test of independence is intended to test how likely it is that an observed distribution of two categorical variables is due to chance. Its robustness with respect to distribution of the data, applicability in studies where parametric assumptions remain unmet, and flexibility to deal with two or multiple groups have widened its application in scientific literature. In our bivariate analyses, we aimed to examine the association between diabetes prevalence, awareness, treatment, and control and the selected background characteristics that included more than one nominal variable. So, we preferred the chi-squared test. However, small sample size, fewer than required cell frequency and large number of categories may weaken the chi-squared test and provide biased significance. Our analysis is free of all these limitations. We haven't used the Cramer's V to assess the strength of the significant chi-square results as we included all the explanatory variables in the multivariate regression analysis (i.e., following our conceptual framework) irrespective of the corresponding significance of the chi-squared tests [29].

To examine the role of socioeconomic status affecting diabetes, a multivariate logistic regression model was used with the following regression specification:

$$logit(Y_i) = \ln\left(\frac{\pi}{1-\pi}\right) = \alpha + \gamma_i S_i + \beta_i X_i + \varepsilon_i \tag{1}$$

Where $Y_i$ is the outcome of interest, which is a dichotomous variable with the value equal to 1 if the individual $i$ has diabetes (0 otherwise); $\pi$ is the probability of having diabetes for individual $i$ (i.e., $Y_i = 1$); $S_i$ is the socioeconomic status of individual $i$; $X_i$ is a vector of relevant socio-demographic covariates that influence the outcome. Following the model specification in Eq (1), factors affecting diabetes prevalence, awareness, treatment, and control were examined by estimating the adjusted odds ratio (aOR) from separate multivariate logistic regression models. Stata v.16 was used for analysis, and appropriate sampling weights for BDHS 2017–18 were applied using Stata's survey estimation procedures ("svy" command) to get nationally representative estimates of the adult population of Bangladesh after adjusting for sample strata and clusters.

## Ethics approval

Demographic and Health Survey Data collection procedures for the 2017–18 BDHS were approved by the Institutional Review Boards of the ICF International, Rockville, MD, USA and Bangladesh Medical Research Council, Dhaka, Bangladesh. Informed consent was obtained from all respondents in the survey before asking questions, and separately before obtaining biomarker and anthropometric measurements. Respondents who did not provide consent were excluded from the analysis for the current study.

## Results

### General characteristics of study participants

Table 1 shows the characteristics of the sampled respondents aged 18 years and above in the 2017–18 BDHS. Our study sample comprised 45% young adults aged 18–34 years, 57%

**Table 1. Characteristics of the study population, Bangladesh 2017–18.**

| Background characteristics | Number | Percentage |
|---|---|---|
| **Age (in years)** | | |
| 18–34 | 5418 | 44.8 |
| 35–44 | 2427 | 20.1 |
| 45–54 | 1680 | 13.9 |
| 55–64 | 1362 | 11.3 |
| 65+ | 1204 | 10.0 |
| **Sex** | | |
| Male | 5179 | 42.8 |
| Female | 6913 | 57.2 |
| **Educational attainment** | | |
| No education | 3116 | 25.8 |
| Primary | 3618 | 29.9 |
| Secondary incomplete | 3136 | 25.9 |
| Secondary complete or higher | 2223 | 18.4 |
| **Wealth index** | | |
| Low | 4729 | 39.1 |
| Middle | 2492 | 20.6 |
| High | 4870 | 40.3 |
| **Locality** | | |
| Urban | 3208 | 26.5 |
| Rural | 8883 | 73.5 |
| **Body mass index (BMI)** | | |
| Underweight | 2068 | 17.1 |
| Normal | 7013 | 58.0 |
| Overweight/obese | 2878 | 23.8 |
| Unknown[a] | 132 | 1.1 |
| **Hypertensive** | | |
| No | 8753 | 72.4 |
| Yes | 3338 | 27.6 |
| **Total** | **12092** | **100.0** |

Note: [a] Either height or weight or both were missing.

females, and 74% rural residents. Nearly a quarter of the sample was overweight or obese, and 28% were hypertensive.

## Diabetes among Bangladeshi adults in 2017–18

Approximately one in ten (9.9%) adults aged 18 years or above were diagnosed to have diabetes in Bangladesh. Between age groups 18–34 and 35–44 years, the prevalence of diabetes more than doubled from 5.3% to 11.1% and was higher among the older age groups. Surprisingly, we observed a slight reduction (from 15.8% to 15.1%) in diabetes prevalence among those aged 65 years or above compared to their younger age group of 55–64 years. Females showed a slightly higher prevalence of diabetes than males except for the age group 65 years or more (see Fig 1).

For our bivariate analyses, we carried out chi-squared tests to determine whether there was an association between the selected explanatory variables based on the conceptual framework and diabetes prevalence, awareness, treatment, and control. Given the multiple numbers of

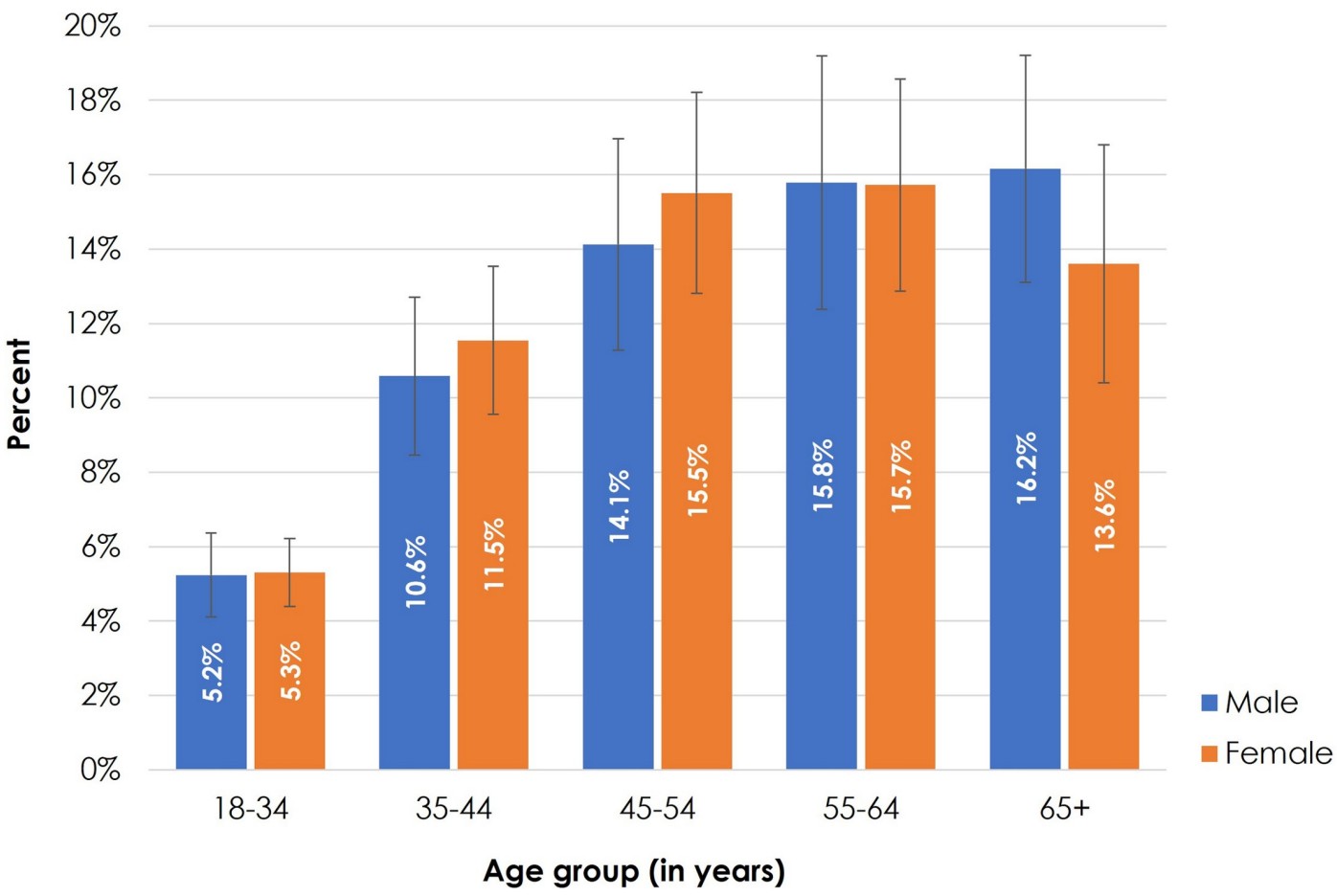

**Fig 1. Prevalence of diabetes by age group and sex in Bangladesh, 2017–18.**

tests performed under bivariate analyses, we estimated the Bonferroni corrected p-value to be 0.0011, meaning p-values over this threshold are not significant [30]. Considering the Bonferroni-corrected p-value threshold, having diabetes was significantly positively associated with age, socioeconomic status, BMI, hypertensive status, and urban residence. While there was no evidence of association with gender or educational attainment in bivariate analyses (see Table 2), only 6% of the people from low socioeconomic status had diabetes, which increased to 15% for the high socioeconomic status (p<0.001). The prevalence of diabetes was around 8% among people living in rural areas, who were normal/underweight, and was non-hypertensive. Prevalence was nearly double among overweight/obese people and hypertensives (p<0.001). From the bivariate analyses, we also found that among the adults with diabetes, awareness and taking medication were significantly associated with age, socioeconomic status, BMI, and hypertensive status (p<0.001). Among the adults with diabetes, having their diabetic condition under control was significantly associated only with age (p<0.001) and hypertensive status (p = 0.0011).

Close to 40% of adults with diabetes are aware of the condition, 35% take medication, and 11% have their diabetic condition under control. In other words, while more than 90% of the respondents with diabetes who are aware of their condition take required medications, fewer than one in three are able to keep their condition under control (see Fig 2). A simple simulation, based on these estimates, indicates increasing awareness to 100% among the people with

**Table 2. Prevalence, awareness, and management of diabetes among the adult population by background characteristics, Bangladesh 2017–18.**

| Background characteristics | Prevalence of diabetes | | Among the respondents with diabetes who are: | | | | | |
| --- | --- | --- | --- | --- | --- | --- | --- | --- |
| | | | Aware of condition | | Taking medications | | Controlled blood glucose level | |
| | Percent | $\chi^2$ test p-value | Percent | $\chi^2$ test p-value | Percent | $\chi^2$ test p-value | Percent | $\chi^2$ test p-value |
| **Age (years)** | | | | | | | | |
| 18–34 | 5.3 | <0.001 | 14.5 | <0.001 | 12.5 | <0.001 | 2.8 | <0.001 |
| 35–44 | 11.1 | | 32.2 | | 29.6 | | 7.2 | |
| 45–54 | 14.9 | | 48.5 | | 44.0 | | 14.5 | |
| 55–64 | 15.8 | | 55.7 | | 51.8 | | 15.3 | |
| 65+ | 15.1 | | 51.6 | | 46.2 | | 18.4 | |
| **Sex** | | | | | | | | |
| Male | 10.5 | 0.072 | 37.1 | 0.443 | 33.3 | 0.337 | 10.8 | 0.995 |
| Female | 9.5 | | 39.7 | | 36.4 | | 10.8 | |
| **Education** | | | | | | | | |
| No education | 9.9 | 0.274 | 40.6 | 0.007 | 37.3 | 0.006 | 12.7 | 0.108 |
| Primary | 10.2 | | 30.5 | | 27.2 | | 10.5 | |
| Secondary incomplete | 9.1 | | 41.8 | | 36.6 | | 6.9 | |
| Secondary complete or higher | 10.9 | | 44.4 | | 42.1 | | 13.3 | |
| **Socioeconomic status** | | | | | | | | |
| Low | 5.9 | <0.001 | 26.8 | <0.001 | 25.4 | 0.001 | 10.7 | 0.827 |
| Middle | 8.0 | | 36.6 | | 31.4 | | 12.2 | |
| High | 14.9 | | 43.5 | | 39.7 | | 10.5 | |
| **Locality** | | | | | | | | |
| Urban | 13.3 | <0.001 | 37.0 | 0.468 | 33.9 | 0.600 | 9.3 | 0.237 |
| Rural | 8.7 | | 39.4 | | 35.6 | | 11.6 | |
| **BMI** | | | | | | | | |
| Underweight | 6.2 | <0.001 | 19.1 | <0.001 | 19.1 | <0.001 | 11.0 | 0.906 |
| Normal | 8.6 | | 38.5 | | 34.8 | | 10.9 | |
| Overweight/obese | 15.6 | | 44.0 | | 39.9 | | 10.4 | |
| Not measured | 14.8 | | 42.0 | | 32.9 | | 15.3 | |
| **Hypertension status** | | | | | | | | |
| Not hypertensive | 7.4 | <0.001 | 26.3 | <0.001 | 23.4 | <0.001 | 8.0 | 0.0011 |
| Hypertensive | 16.6 | | 53.0 | | 48.8 | | 14.2 | |
| **Total** | **9.9** | - | **38.5** | - | **35.0** | - | **10.8** | - |
| **Number of respondents** | 12092 | | 1201 | | 1201 | | 1201 | |

Note: Bonferroni-corrected p-value threshold for multiple testing is 0.0011

diabetes would increase the use of medication to 91% from currently 35%. This may improve blood glucose control to 28% from currently 11%. The analyses assumed the medication status, including the provider and the client compliances, remained unchanged.

Similar to prevalence, awareness of diabetes and medication among the respondents with diabetes increased with age and higher socioeconomic status (see Table 2). Male-female and urban-rural differences were small and not statistically significant. Levels of education, BMI, and hypertensive status were significantly associated with awareness of the condition and taking medications. Except for age and hypertensive status, no other background characteristics were significantly associated with blood glucose control among the respondents with diabetes. Older and hypertensive individuals appeared to have better control of their blood glucose levels.

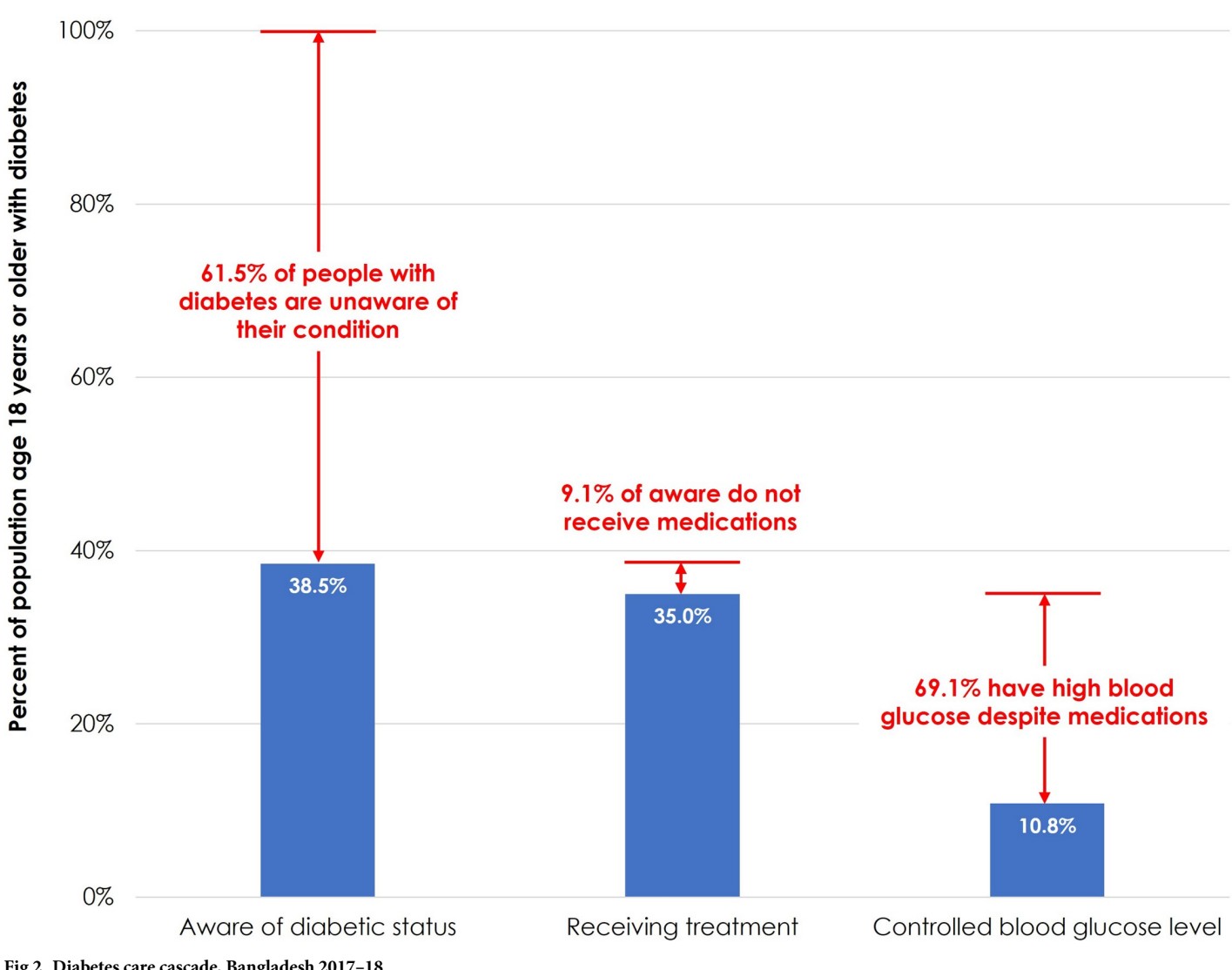

**Fig 2. Diabetes care cascade, Bangladesh 2017–18.**

## Correlates of diabetes among Bangladeshi adults

In a multivariate logistic regression model, an adjusted odds ratio (aOR) denotes an odds ratio that has been adjusted to account for other explanatory variables. In this study, we included seven explanatory variables in the logistic regression following the conceptual framework. Therefore, the corresponding odds ratio of one variable is adjusted for the other six variables, and we address that as an aOR. We used the aORs as a tool to examine the association of diabetes with selected socioeconomic, demographic, and health status-related factors (i.e., correlates), and followed standard practices for its mathematical specification and explanation [31, 32]. Correlates of diabetes that were statistically significant in bivariate analyses remained significant even after adjusting for background characteristics (see Table 3). There was a clear socioeconomic gradient for diabetes prevalence—compared to the low socioecnomic status, the odds of having diabetes was 1.3 and 2.4 times higher for people from the middle and the high socioeconomic status (p<0.05 and p<0.001, respectively). The odds of having diabetes for the youngest age group (18–34 years) was half than their older age group coungerparts (35

**Table 3. Adjusted odds ratios of diabetes prevalence, awareness, and management among the adult population, Bangladesh 2017–18.**

| Background characteristics | Has diabetes | | | Aware of condition | | | Taking medications | | | Controlled blood glucose level | | |
|---|---|---|---|---|---|---|---|---|---|---|---|---|
| | aOR | p-value | 95% CI | aOR | p-value | 95% CI | aOR | p-value | 95% CI | aOR | p-value | 95% CI |
| **Age** *(reference category: 35–44 years)* | | | | | | | | | | | | |
| 18–34 | 0.48 | <0.001 | 0.39–0.59 | 0.32 | <0.001 | 0.20–0.51 | 0.31 | <0.001 | 0.19–0.50 | 0.36 | 0.018 | 0.16–0.84 |
| 45–54 | 1.40 | 0.002 | 1.13–1.74 | 1.92 | 0.002 | 1.27–2.91 | 1.77 | 0.006 | 1.17–2.66 | 2.16 | 0.019 | 1.14–4.11 |
| 55–64 | 1.60 | <0.001 | 1.27–2.01 | 2.59 | <0.001 | 1.65–4.06 | 2.51 | <0.001 | 1.59–3.94 | 2.41 | 0.008 | 1.25–4.65 |
| 65+ | 1.57 | 0.001 | 1.20–2.05 | 2.62 | <0.001 | 1.61–4.27 | 2.31 | 0.001 | 1.41–3.78 | 2.98 | 0.002 | 1.48–6.02 |
| **Sex** *(reference category: male)* | | | | | | | | | | | | |
| Female | 0.97 | 0.622 | 0.84–1.11 | 1.53 | 0.008 | 1.12–2.10 | 1.59 | 0.004 | 1.16–2.19 | 1.36 | 0.180 | 0.87–2.13 |
| **Education** *(reference category: no education)* | | | | | | | | | | | | |
| Primary | 1.31 | 0.006 | 1.08–1.59 | 0.93 | 0.728 | 0.60–1.42 | 0.91 | 0.682 | 0.58–1.42 | 1.25 | 0.432 | 0.71–2.20 |
| Secondary incomplete | 1.20 | 0.119 | 0.96–1.50 | 1.92 | 0.007 | 1.20–3.06 | 1.73 | 0.026 | 1.07–2.78 | 1.03 | 0.939 | 0.53–1.97 |
| Secondary complete+ | 1.23 | 0.122 | 0.95–1.59 | 2.12 | 0.002 | 1.32–3.43 | 2.29 | 0.001 | 1.41–3.71 | 2.18 | 0.023 | 1.12–4.27 |
| **Socioeconomic status** *(reference category: low)* | | | | | | | | | | | | |
| Middle | 1.29 | 0.034 | 1.02–1.64 | 1.34 | 0.249 | 0.81–2.20 | 1.11 | 0.660 | 0.68–1.84 | 1.04 | 0.910 | 0.54–2.01 |
| High | 2.35 | <0.001 | 1.90–2.90 | 1.60 | 0.031 | 1.04–2.45 | 1.44 | 0.092 | 0.94–2.20 | 0.81 | 0.410 | 0.48–1.34 |
| **Locality** *(reference category: urban)* | | | | | | | | | | | | |
| Rural | 0.86 | 0.093 | 0.73–1.03 | 1.28 | 0.123 | 0.94–1.76 | 1.24 | 0.180 | 0.91–1.69 | 1.11 | 0.619 | 0.72–1.73 |
| **BMI** *(reference category: normal)* | | | | | | | | | | | | |
| Underweight | 0.79 | 0.053 | 0.62–1.00 | 0.42 | 0.009 | 0/22-0.81 | 0.52 | 0.044 | 0.27–0.98 | 1.11 | 0.798 | 0.51–2.40 |
| Overweight/obese | 1.47 | <0.001 | 1.25–1.71 | 0.96 | 0.797 | 0.69–1.33 | 0.94 | 0.702 | 0.68–1.29 | 0.90 | 0.658 | 0.57–1.42 |
| Not measured | 1.15 | 0.583 | 0.70–1.91 | 0.57 | 0.231 | 0.23–1.43 | 0.46 | 0.084 | 0.19–1.11 | 0.94 | 0.909 | 0.31–2.82 |
| **Hypertensive status** *(reference category: not hypertensive)* | | | | | | | | | | | | |
| Hypertensive | 1.57 | <0.001 | 1.35–1.82 | 1.95 | <0.001 | 1.44–2.63 | 2.00 | <0.001 | 1.47–2.72 | 1.34 | 0.149 | 0.89–2.01 |
| **Constant** | 0.05 | <0.001 | 0.04–0.08 | 0.14 | <0.001 | 0.07–0.26 | 0.13 | <0.001 | 0.07–0.25 | 0.05 | <0.001 | 0.02–0.12 |
| **Number of respondents** | 12092 | | | 1201 | | | 1201 | | | 1201 | | |

Note: aOR: adjusted odds ratio; CI: confidence interval.

−44 years) (p<0.001). Using the same reference age group, the odds were 1.4 times higher among people age 45−54 years and 1.6 times higher among people age 55 years or above (p<0.05 and p = 0.001, respectively). The odds of overweight/obese people having diabetes was nearly 1.5 times than those which have normal BMI. People with hypertension had nearly 2.6 times higher odds of having diabetes than those who were not (p<0.001). Although the odds of having diabetes was found to be higher with educational attainment compared with no education, the aOR was only significant for primary education in our analysis. Our logistic regression model also indicated the odds of having diabetes was lower among people living in rural areas, but the significance level fell outside the traditional threshold (p = 0.089).

The odds of being aware of diabetes was more strongly associated with socioeconomic status than the odds of taking anti-diabetic medications. The odds of respondents with diabetes from the high socioeconomic status was 1.6 times higher to be aware of their condition than the respondents with diabetes from the low socioecnomic status (p<0.05) (see Table 3). The odds of being aware and taking medications also increased with age. Unlike diabetes prevalence, odds of awareness and medication were found to be gender-dependent, with females having 1.5 and 1.6 times higher odds to be aware of the condition or for taking medications, respectively (p<0.05). As expected, people educated higher than primary levels had higher odds to be aware of their diabetic condition than their counterpart with no education, while the odds of taking medication was two-times higher among those who completed at least

secondary education (p<0.001). When biological correlates were considered, respondents with both hypertension and diabetes had nearly two-times higher odds to be aware and to take medications (p<0.001). Both awareness of and medication for diabetes were independent of urban/rural residence and BMI.

We also found using our multivariable logistic regression model that a person's gender, wealth status, urban/rural residence, BMI, and hypertensive status were not associated with control of diabetes (see Table 3). When adjusted for possible explanatory variables, the odds of having diabetes under control was 1.3 times higher among people with hypertension than non-hypertensives, although the statistical significance was lost (Table 3). Our analysis found older age and higher education attainment (completion of secondary education or more) significantly increased the odds of having diabetes under control (p<0.05).

## Discussion

From the most recent nationally representative survey data, we estimated that 9.9% of the population aged 18 years and above had diabetes in Bangladesh in 2017–18, which is substantially higher than the available country estimates from commonly used global sources such as IDF or WHO [1, 33]. IDF estimated 8.4 million adults in Bangladesh to have diabetes in 2019 [1], whereas our estimated diabetes prevalence translates into 10.8 million people using the 2019 population estimates from the United Nations [34]. Between 2011 and 2017–18, the overall diabetes prevalence among Bangladeshi adults aged 35 years or above increased from 11.0% to 13.7%, which is equivalent to a growth rate of 4.1% annually. It is unusual for a chronic condition to change so rapidly, so we first looked into explaining the rapid rise in diabetes levels in Bangladesh. An analysis of causes of death data from a rural sub-district of Bangladesh over 30 years period indicated Bangladesh was already at the third stage of epidemiological transition from acute, infectious, and parasitic diseases to degenerative, chronic NCDs [35]. At the current stage of epidemiological transition, the burden of major NCDs like diabetes, cardiovascular disease, and cancers is likely to increase in the coming years. Another possible reason for the rapid rise in diabetes levels would be changing patterns of diabetes risk factors (e.g., socioeconomic status and other social determinants of health, age, BMI, co-morbidities like hypertension), which are discussed below.

We found higher socioeconomic status to be a strong determinant of diabetes. This is in line with other studies [15, 16, 22, 23, 36–39], but contradicts the findings from high-income countries where diabetes prevalence was more concentrated among the poor [26, 40]. In our analysis, socioeconomic status played a statistically significant role in predicting the likelihood of an individual having diabetes and influencing an individual's awareness of having the disease. However, the influence of socioeconomic status on the likelihood of receiving anti-diabetic medication only approached significance (p = 0.07), and we found no significant association between socioeconomic status and control of diabetes. A study by Rahman and colleagues on the 2011 BDHS found diabetic individuals belonging to the lower socioeconomic group were less likely to be aware of their condition and less likely to be under anti-diabetic medication primarily due to the high cost of diabetes care [19]. A recent study found that the total annual per capita expenditure on medical care to be 6.1 times higher for people with diabetes than non-diabetics [41]. The increasing prevalence of diabetes and fewer diabetics being aware and taking medication could imply a huge burden of medical care cost nationally, most of which is paid out of pocket.

A systematic review by Grintsova et al. (2014) from high-income countries, where inequalities exist across different healthcare systems, found a trend towards worse healthcare for patients with low socioeconomic status and people with diabetes living in deprived areas less

often achieve glycemic control targets [42]. Our study, however, indicated that people from high socioeconomic status were not significantly better than low socioeconomic status in achieving glycemic control. One possible reason for such a finding could be that lifestyle-related risk factors of chronic diseases are highly prevalent in South Asia, and Bangladesh has the worst risk factor profile among the nations in the region [24, 43, 44]. Among the lifestyle-related factors, diet composition as well as excess energy intake are the main contributing factors—the socioeconomically better-off class and urban residents are increasingly being exposed to processed food, sugar-sweetened beverages, and trans-fats, and there is no clear guideline in the country yet to restrict these contents in the food products [24, 45, 46]. A study on knowledge and self-care practices regarding diabetes found that despite high knowledge level regarding diabetes among the sampled diabetic population, about 90% of respondents did not test their blood glucose regularly or did not follow dietary advice given by diabetes educators, and only about one-third of respondents partially followed rules for portion control before eating [47].

Older age was also found to be a powerful determinant of diabetes among Bangladeshi adults, which is consistent with the findings from recent studies elsewhere [40, 48–52]. Age contributes to the pathophysiology of diabetes, type 2 diabetes in particular, due to the combined effects of increasing insulin resistance and impaired pancreatic function with aging [53]. Since diabetes is a chronic condition and is carried throughout life, about three-quarters of diabetics who have the condition in 2010 would still have it in 2020, when they would be a decade older. In addition, a further two million new cases would have been added each year in Bangladesh during that decade. With declining fertility and a steady increase in life expectancy in Bangladesh (between 2010 and 2019, life expectancy at birth for males increased from 67 to 71 years and for females from 69 to 74 years) [54], we may expect to see 'accumulation' of the number of people with diabetes to continue in the coming years. Older age was also found to be a statistically significant determinant for increasing awareness, treatment initiation, and control of diabetes. Routine health screening recommendations explicitly incorporate older adults (viz. middle-aged and above) to be screened for diabetes in most countries [55]. Bangladesh's National NCD Program currently uses an age threshold of 40 years or older (and any pregnant women) for initial screening for diabetes at a primary-level health facility (community clinics and Upazila health complexes—UHCs) for any service contact, which may have increased the likelihood of diagnosis and treatment at an older age. Survival bias may also be a reason for a higher level of diagnosis at older ages as individuals who have uncontrolled diabetes or complications related to diabetes are more likely to die at a younger age compared to individuals who had a late onset of the condition and/or controlled it well, [56] making the latter groups over-represented in large, population-based cross-sectional surveys. Though diabetes is a progressive condition and blood glucose levels are known to increase with age, individuals diagnosed with type 2 diabetes at younger ages and being diabetic for a longer duration may, in the long run, have a more severe form of the disease and find it more difficult to achieve glycemic control with current treatment modalities [57]. This phenomenon may explain why some older individuals have significantly better control of diabetes compared to some younger adults as they may have late onset of diabetes.

Excess weight is associated with an increased prevalence of diabetes, hypertension, and certain cancers. In addition, both obesity and diabetes are associated with an increased risk for cardiovascular disease-related mortality and other mortalities [58]. A recent study found that the chance of developing diabetes among overweight adult persons was double that of their normal-weight counterparts in Bangladesh [59]. An important finding for Bangladesh from this study is that overweight and/or obese persons are more likely to have diabetes but not more likely to be aware, taking medications, or controlling their blood glucose levels.

It was surprising that the place of residence (urban/rural) was not significantly associated with the diabetes outcomes used in our study, after controlling for relevant covariates. In many other studies, diabetes prevalence tends to be higher in urban areas, even in Bangladesh [23, 38, 39]. Though the lifestyle is expected to be less sedentary in rural areas than in urban cities, overall socioeconomic improvement and high consumption of rice may partly explain the narrowing gap in diabetes prevalence in rural and urban areas in recent years. Studies found that the most popular, high yielding rice varieties in Bangladesh have a very high glycemic index [60, 61]. As expected, higher educational attainment was found to be significantly associated with diabetes awareness, treatment, and control of blood sugar levels.

Our findings from the diabetes care cascade (see Fig 2) revealed that 62% of the people with diabetes are unaware that they have diabetes. This level of undiagnosed diabetes is higher than the regional average (57%) or for middle-income countries (53%) and more in line with low-income countries 67% [1]. The findings also showed that though the gap to treatment initiation once diagnosed is small (i.e., 91% of diagnosed people having anti-diabetic medication), only a third of them were able to reach the recommended lower blood glucose level. Overall, only one-in-nine diabetic persons were able to control their diabetes in 2017–18, which is slightly lower than the 2011 level (12.5%). Given the current status of diabetes control, even if we were able to increase awareness to 100% among people with diabetes, only 28% would be able to achieve the required blood glucose level. This calls for immediate attention to the existing health service delivery mechanism for diabetes in the country. The most recent health facility survey in Bangladesh indicated 53% of health facilities in Bangladesh offered services for diabetes (that is, providers in the facility diagnose, manage, and prescribe treatment for people with diabetes) in 2017, a substantial increase from 18% in 2014 [62]. A health facility was classified as "ready" to provide services for a specific NCD if all the following components were present: relevant guidelines; at least one trained provider; necessary equipment, diagnostic supplies, and medicines [63]. Using this globally acceptable standard of service readiness, the 2017 Bangladesh Health Facility Survey indicated that none of the existing health facilities have all components present to deliver diabetes services—only 17% of facilities had guidelines for diagnosis and management of diabetes available on the day of the survey; 33% had staff ever trained in diabetes; 54% had blood glucose testing capacity, while 46% had urine protein and urine glucose testing capacity; 40% of facilities had metformin, and 29% had injectable insulin, 9% had glibenclamide, and 19% had injectable glucose solution available [62]. Low readiness for diabetes services was found in other studies based on Bangladesh as well [64–66].

In summary, our analyses clearly indicated that adults from higher socioeconomic status were significantly more likely than those from low socioeconomic status to become diabetic, be aware of their condition, and be under medication, but not significantly better in achieving glycemic control. With Bangladesh's ongoing economic progress and demographic transition, we may expect to see an 'accumulation' of the number of people with diabetes to continue in the coming years. Given the increasing number of diabetes cases in the country and low level of awareness as well as glycemic control among the people with diabetes, the traditional physician-centric healthcare delivery system in Bangladesh appears to be inadequate to meet the demands for diabetes treatment and management [67]. In order to design effective diabetes screening and management services to maximize NCD coverage in the future, the following policy and programmatic interventions can be considered:

1. **Expand coverage of facility-based services for diabetes in all sectors:** The government's policy on screening and management of diabetes has not been consistent over the years. Under the national health sector program for 2011–2016, 300 UHCs and corresponding community clinics provided diabetes (and hypertension) screening [68]. In the subsequent

health sector program for 2017–22, however, the government moved away from this screening initiative and started piloting an NCD management model in 62 out of 492 upazilas (sub-districts) to provide comprehensive screening, referral, and management services. The government needs to scale up the piloting without delay and initiate screening in the rest of the primary-level public facilities (i.e., 13,442 community clinics, 420 UHCs, 98 mother and child welfare centers, 4,011 union health and family welfare centers, 1,275 union sub-centers, and other sub-district/union-level hospitals) to expand the service coverage for diabetes [62]. Since the expected number of people with diabetes in the country cannot be handled by the public sector alone, the government needs to explore alternative options and take advantage of the presence of private sector and non-governmental organizations (NGOs) to train non-physicians to conduct screening and follow-up examinations to handle non-complicated/non-critical cases with appropriate referral chain guidelines. While the expansion of service coverage through the public sector will largely cover the rural areas, it would be more resource-efficient to engage the private and NGO sectors to cover the urban cities.

2. **Revise the existing screening and management modalities:** The age threshold for the initial NCD screening at primary-level health facilities and referral as per the National NCD Program protocol need to be reduced to 35 years (from 40 years) to expand screening coverage for both diabetes and hypertension. Our analysis indicates that diabetes prevalence doubled after age 35 years, and the proposed change in age threshold for screening will be able to identify the people with diabetes earlier for better treatment. In collaboration with the government's NCD Program, BADAS recently finalized a comprehensive Diabetes Care Guideline [69], which needs to be implemented nationwide to ensure a consistent, integrated approach for diabetes screening, treatment, and management in the country. The government may also explore opportunities for public-private partnerships to nationally scale up BADAS' Health Care Development Program, which is currently piloting a model of integrated care service delivery in urban and rural areas with a focus on the integrated healthcare approach for diabetes and other major NCDs [70].

3. **Improve health system's capacity for diabetes management:** The low readiness of health facilities for diabetes management as shown by the 2017 Bangladesh Health Facility Survey can be addressed by training health providers in diabetes diagnosis and management, making available guidelines and protocols, increasing diagnostic and laboratory capacities, and ensuring the stock of medicines such as metformin, insulin, and glibenclamide. The planning and procurement capacities of the National NCD Program also need to be improved to ensure an uninterrupted supply of drugs, equipment, and laboratory reagents to the health facilities. Under the ongoing health sector program, there are provisions for program capacity development through technical assistance, which can be utilized by the National NCD Program.

4. **Explore innovative approaches to improve awareness and medication compliance:** Our analyses indicated that higher socioeconomic status was positively associated with diabetes prevalence, awareness, and medication but not with achieving glycemic control. To improve knowledge and medication compliance among the better-off individuals, innovative measures utilizing mobile and information technology can be explored. A randomized controlled trial in Bangladesh showed that in people with type 2 diabetes attending a specialized hospital, improvement of glycemic control could be achieved by mobile phone text messaging [71]. Public health campaigns alone, while they can increase awareness, have not proven effective in preventing diabetes [72]. Use of social media and mobile phone text

messaging may, therefore, provide efficient and cost-effective approaches to increase diabetes knowledge, improve self-care and management, and enhance healthcare seeking behavior among the people with diabetes.

5. **Strengthen the government's stewardship role to implement multisectoral actions:** Diabetes prevention and control require strong national stewardship of the government's Ministry of Health and Family Welfare (MOHFW) to coordinate a meaningful multisectoral response. In 2018, the MOHFW adopted a multisectoral action plan for prevention and control of NCDs that involved 30 ministries and agencies [73]. The MOHFW needs to strengthen its stewardship capacity to coordinate interaction with other sectors and agencies to implement the following actions as priority: formulate guidelines to restrict sugar and trans-fat contents in the food products (with the ministries of Food, Commerce, and Industries); promote healthy diets by switching from very high glycemic index rice varieties (with the ministries of Agriculture, Food, and Commerce); and promote healthy lifestyle (with the ministries of Education, Food, Local Government, Housing and Public Works, and Youth and Sports).

## Strengths and limitations

Our study provides evidence of the correlates of diabetes using the most recent nationally representative sample. To our knowledge, this is the first survey in Bangladesh, which collected information on diabetes for the adult (i.e., 18 years and older) population by using World Health Organization recommended guidelines. In this analysis, we accounted for the sample stratification and clustering in the BDHS so that the findings are truly representative of the national status in terms of diabetes estimates.

Unlike the main module of the BDHS questionnaire, where the response rate was 99%, the non-response rate for the biomarker questionnaire was high—16.3% of the respondents either were not available or refused to participate in blood glucose measurements, yielding a response rate of 83.7%. We performed an empirical diagnostic check to ensure that our final sample continues to be representative of the total sample, and consequently, the country population (see S1 Table in S1 File). Lastly, multivariable regression analyses in this study were conducted on a single round of cross-sectional data (i.e., BDHS 2017–18) separately for diabetes prevalence, awareness, treatment, and control, and no post-hoc analysis like Bonferroni correction for multiple-hypothesis testing was performed.

## Conclusion

Comparable estimates of diabetes prevalence among older adults in Bangladesh indicate a steady rise in diabetes in recent years, which would hinder achieving the health-related Sustainable Development Goals by 2030. The rising prevalence of diabetes is only the tip of the iceberg, while the huge number of people with diabetes being unaware of their uncontrolled blood glucose level leading to increased morbidity and mortality could be the real threat. Immediate measures to increase screening coverage and exploration of poor control of diabetes is required to mitigate the situation. The involvement of private sector and NGOs, coupled with innovative measures utilizing mobile and information technology, should also be explored to provide support for people with diabetes in Bangladesh.

## Supporting information

**S1 File.**
(DOCX)

## Acknowledgments

icddr,b acknowledges with gratitude the commitment of USAID to its research efforts. icddr,b is also grateful to the Governments of Bangladesh, Canada, Sweden, and the UK for providing core/unrestricted support. We also acknowledge the institutions and individuals who had made the 2017–18 Bangladesh Demographic and Health Survey possible. This work could not have been done without the full support of the BDHS Technical Working Group (TWG), which includes representatives from NIPORT, The DHS Program, MEASURE Evaluation, Data for Impact (D4I), icddr,b, and USAID/Bangladesh for designing and implementing the surveys. The 2017–18 BDHS data were collected and processed by Mitra & Associates. Lastly, the authors would like to thank the participants of the Bangladesh Demographic and Health Surveys who participated in this study.

## Author Contributions

**Conceptualization:** Karar Zunaid Ahsan, Kanta Jamil, Peter Kim Streatfield.

**Data curation:** Nitai Chakraborty.

**Formal analysis:** Karar Zunaid Ahsan, Afrin Iqbal, Nitai Chakraborty.

**Methodology:** Karar Zunaid Ahsan, Kanta Jamil, M. Moinuddin Haider.

**Project administration:** Shusmita Hossain Khan.

**Supervision:** Kanta Jamil, Peter Kim Streatfield.

**Validation:** M. Moinuddin Haider.

**Writing – original draft:** Karar Zunaid Ahsan, Afrin Iqbal, Peter Kim Streatfield.

**Writing – review & editing:** Karar Zunaid Ahsan, Afrin Iqbal, Kanta Jamil, M. Moinuddin Haider, Shusmita Hossain Khan, Nitai Chakraborty, Peter Kim Streatfield.

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
