## [Decision Letter · Decision Letter 0]

8 Jul 2021

PONE-D-21-02392

Socioeconomic disparities in diabetes prevalence and management among the adult population in Bangladesh

PLOS ONE

Dear Dr. Ahsan,

Thank you for submitting your manuscript to PLOS ONE. Now I have received the review reports from two distinguished reviewers. I myself have gone through the manuscript meticulously with great interest. After careful consideration, we feel that it has merit but does not fully meet PLOS ONE’s publication criteria as it currently stands. Therefore, we invite you to submit a revised version of the manuscript that addresses the points raised during the review process. Please see my comments below in addition to the reviewers concern.

The manuscript has lost its focus; it has failed to comply with the title. It its current format, the manuscript looks like a monograph though PLOS ONE has not problem with space. I do not see any justification for keeping the following sections in the manuscript: Correlates of diabetes among women of reproductive age; and Changes in correlates of diabetes between 2011 and 2017-18 among older adults. Please consider removing these two sections to make a focused manuscript.The tables should provide the 'n' in any format to demonstrate the sample size f different outcome variables.

We look forward to receiving your revised manuscript.

Kind regards,

Mohammad Bellal Hossain

Academic Editor

PLOS ONE

Journal Requirements:

Reviewers' comments:

Reviewer's Responses to Questions

**Comments to the Author**

1. Is the manuscript technically sound, and do the data support the conclusions?

Reviewer #1: Partly

Reviewer #2: Yes

2. Has the statistical analysis been performed appropriately and rigorously? 

Reviewer #1: No

Reviewer #2: Yes

3. Have the authors made all data underlying the findings in their manuscript fully available?

Reviewer #1: No

Reviewer #2: Yes

4. Is the manuscript presented in an intelligible fashion and written in standard English?

Reviewer #1: No

Reviewer #2: Yes

5. Review Comments to the Author

Reviewer #1: Thanks authors for analysing the latest data from the Bangladesh Demographic and Health Survey (BDHS) to explore socioeconomic disparities in diabetes prevalence and management among the adult population in Bangladesh. This study is important to understand the growing burden of diabetes in Bangladesh. However, this manuscript does need major revisions with clarification, particularly in the methods and analysis section. I do have few questions and changes to suggest:

Major comments:

- As suggested by the IDF, we cannot label respondents as diabetic patients. It should be, ‘people with diabetes’. Please change throughout the manuscript.

- The current title is not consistent with the way the authors analyse and reported data. According to the title, the authors should consider socioeconomic status (SES) as exposure and adjust all the confounders rather than putting all of them in kitchen sink regression model.

- The manuscript is written poorly with a lack of clear direction. For example, the focus of the manuscript is on the socioeconomic disparities in diabetes prevalence, however, most of the literature are not in relation to the issue. Secondly, the authors defined type2, type1, GDM and provided every detail of global prevalence which are not needed, while diabetes management, an important part of the manuscript, is completely ignored.

- It is not appropriate to investigate the changes of risk factors, especially for time variants factors e.g. age, BMI, and hypertension status, when they modelled each survey data separately.

- The BMI should be categorised based on Asian cut-off as suggested by the WHO expert consultation due to the high risk of type 2 diabetes and cardiovascular disease in Asian people at lower BMIs than the existing WHO cut-off. However, it is not clear whether the authors used Asian or WHO cut-off. About ~17 % of the study population is underweight that has differential impact on diabetes. So, I suggest not to combine underweight and normal weight.

- It is not appropriate to combine wealth quintiles.

- It is unclear whether the authors checked multi-collinearity, interaction or effect modification in the data.

- When the overall prevalence of diabetes is >10% (especially for table 5). For such a common outcome, logistic regression (odds ratio) overestimates or jeopardizes the association. So, the adjusted association (Odds ratio) might be just by chance, not the true association. Prevalence ratio is the suitable option for a cross-sectional study with a common outcome, while the odds ratio is the best option for a case-control study or the rare outcome. Authors can consult with log-binomial or other advanced methods for calculating adjusted prevalence ratio.

- Please provide flowchart of analytic sample. The final report shows that total sample was 12099 (table 13.5.1: women 6918, table13.5.4:men 5181), including pregnant women. However, your analytic sample is 12092 which is only 7 women less compared to the national report.

- For subgroup analysis of women, number of pregnancies/parity should be adjusted in the model.

- For table 2, please report overall numerator and denominator.

- Results should be reported precisely. Prevalence should be reported with 95%CI both in the table and in the text.

- The discussion is sloppy. Policy suggestions for diabetes management need to be compact as currently reported as literature review.

- The manuscript must be edited by a native speaker.

Reviewer #2: 1) General comments and summary of recommendation

Describe your overall impressions of the submission, how it fits within the scope of the journal, and your recommendation.

This is an interesting and worthwhile analysis of an important issue in countries such as Bangladesh. The paper focus on sociodemographic disparities and not just socioeconomic.

General comments would be to talk about people with diabetes rather than diabetics. I understand from clinical colleagues that there is a move away from labelling people based on their disease status.

2) Content:

Please consider the originality, relevance and rigour of the submission and the author’s depth of understanding of the issues being researched. Please comment on the adequacy of the author’s referencing and whether or not the existing knowledge base has been explored and built upon.

Please examine the methodologies used and their appropriateness, and the author’s use of the evidential base. Is all statistical analysis sound? Does it need to be checked further by a statistical expert? Does the conclusion reflect the argument in the main body text, is it supported by factual data, and does it bring something new to the debate? Overall, is the submission scientifically sound?

This is a well thought through paper on an important issue. I have some concerns that should be addressed before the paper is ready for publication.

Major

1. The sample size is small for doing multiple comparisons, and this should be acknowledged.

2. Please give exact p-values unless p<0.0001.

3. There are probably >100 p-values in this paper and so it is inappropriate to focus on any <0.05.

4. Have the authors considered doing a global test for heterogeneity in their tables?

5. On a similar note, it would be more appropriate to calculate tests for trend for factors that show a trend (eg SES, education, etc).

6. Please describe ORs consistently. On line 278, the OR is described as being 31% higher and 2.4 times higher. We do not normally refer to ORs in terms of percentages. Similarly, on line 258 they are referred to as likelihood.

7. I would prefer to

Minor

1. In the Introduction, line 107, please talk about a decline in the probability of dying prematurely (since the probability of dying=1 in all countries).

2. Line 193: please clarify that ‘hypertensive’ refers to measured hypertensive and so there might be some participants with a diagnosis of hypertension who are controlled by medication. This is fine, but it should be clarified.

3. Table 3: BMI is not nutritional status, please re-label.

4. I would prefer to see SES defined as low, middle, high rather than poor, middle and rich.

3) Structure and argument:

Do the Highlights capture the novel points of interest of the paper? Does the abstract summarise the arguments in a succinct and accurate way? Is the manuscript logically structured and do the arguments flow coherently? Does the introduction signpost the arguments and is there enough reference to methodology in the introduction to substantiate the arguments? Do the Material and Methods provide sufficient details to allow the work to be reproduced by an independent researcher? Are the results clear and concise? Please indicate if there is redundancy of information given in text and tables/figures that can be eliminated. Does the conclusion adequately summarise the main results?

All OK.

4) Figures/tables:

Please comment on the author’s use of tables, charts, figures or maps – their relevance in terms of illustrating the arguments and supporting the evidential base, the quality of the formatting and presentation.

The tables require some work. The use of * to indicate p<0.05, <0.001, etc is inappropriate (as already mentioned). There should be tests for trend and it would be better to use floating absolute risks (Easton et al, 1991, Plummer et al 2004) so that appropriate uncertainty is allocated to the reference group and therefore comparisons can be made between any 2 groups and not just with the reference group.

The figures are inadequate.

Figure 1 needs N and Cis so that the reader can determine how reliably each prevalence has been determined.

Figure 2: the y-axis should be ‘% of those with diabetes’. The current tile implies that 100% of the population aged 18 years or older has diabetes.

5) Language:

Is the text well written (i.e. clear to read) and jargon free? Please comment on the quality of English and need for grammatical improvement.

The language and grammar are adequate.

6) Ethical approval:

If humans or animals have been used as research subjects, are statements of ethical approval by a relevant authority present? Where humans have participated in research, informed consent should also be declared. If not present, please detail where you think a further ethics approval/statement is required.

n/a

7) Data availability:

Has data used in the study been adequately described and made available? Is the data curated in a usable format? Is there a 'Data Availability Statement' prior to the reference list providing information on how to access the data, or information on why it has not been made public?

The analyses use publicly available data.

6. PLOS authors have the option to publish the peer review history of their article (what does this mean?). If published, this will include your full peer review and any attached files.

Reviewer #1: No

Reviewer #2: No

---

## [Author Response · Author response to Decision Letter 0]

22 Aug 2021

Comments from the Editor

1. The manuscript has lost its focus; it has failed to comply with the title. It its current format, the manuscript looks like a monograph though PLOS ONE has not problem with space. I do not see any justification for keeping the following sections in the manuscript: Correlates of diabetes among women of reproductive age; and Changes in correlates of diabetes between 2011 and 2017-18 among older adults. Please consider removing these two sections to make a focused manuscript.

Authors’ Response: The sections “Correlates of diabetes among women of reproductive age” and “Changes in correlates of diabetes between 2011 and 2017-18 among older adults” are dropped (p.18, line 342 to p.20, line 392 in the revised manuscript with track changes). Abstract, Methods, and Discussion sections have been updated accordingly.

2. The tables should provide the ‘n’ in any format to demonstrate the sample size f different outcome variables.

Authors’ Response: Updated Table 3 (p.16) by adding ‘n.’ 

3. Please ensure that your manuscript meets PLOS ONE’s style requirements, including those for file naming. The PLOS ONE style templates can be found at

Authors’ Response: The manuscript has been reviewed to ensure that it meets PLOS ONE style requirements.

4. Please provide additional details regarding participant consent. In the ethics statement in the Methods and online submission information, please ensure that you have specified what type you obtained (for instance, written or verbal, and if verbal, how it was documented and witnessed). If your study included minors, state whether you obtained consent from parents or guardians. If the need for consent was waived by the ethics committee, please include this information.

Authors’ Response: We added a subsection on ethics approval under the Methods section (p.11, lines 233–239).

Comments from Reviewer 1

1. As suggested by the IDF, we cannot label respondents as diabetic patients. It should be, ‘people with diabetes’. Please change throughout the manuscript.

Authors’ Response: We thank the reviewer for highlighting this important point. We have replaced ‘diabetics/diabetic patients’ with ‘people with diabetes’ throughout the manuscript.

2. The current title is not consistent with the way the authors analyse and reported data. According to the title, the authors should consider socioeconomic status (SES) as exposure and adjust all the confounders rather than putting all of them in kitchen sink regression model.

Authors’ Response: As per the manuscript’s title, we consider SES as exposure and have built the multivariate model in a stepwise approach (viz., first running the bivariate tables, then putting the potential confounders one by one, testing interactions between SES and the cofounders, and using the goodness of fit test to decide on the final list of variables). We believe that the statistical measures and approach used in the analysis are appropriate to achieve the objectives of the study. In addition, the title also aligns with the analysis, results found, and possible explanations of the findings. 

3. The manuscript is written poorly with a lack of clear direction. For example, the focus of the manuscript is on the socioeconomic disparities in diabetes prevalence, however, most of the literature are not in relation to the issue. Secondly, the authors defined type2, type1, GDM and provided every detail of global prevalence which are not needed, while diabetes management, an important part of the manuscript, is completely ignored.

Authors’ Response: We are surprised that the reviewer feels this way. Out of 6 paragraphs in the introduction section, only one paragraph broadly discussed the global context before getting into the country context. Also, diabetes management and its relation to socioeconomic disparity are noted based on the literature review (p.6, lines 123–128), so it was not completely ignored. 

4. It is not appropriate to investigate the changes of risk factors, especially for time variants factors e.g. age, BMI, and hypertension status, when they modelled each survey data separately.

Authors’ Response: Following the Editor’s recommendation, we dropped the section on “Changes in correlates of diabetes between 2011 and 2017-18 among older adults.”

5. The BMI should be categorised based on Asian cutoff as suggested by the WHO expert consultation due to the high risk of type 2 diabetes and cardiovascular disease in Asian people at lower BMIs than the existing WHO cutoff. However, it is not clear whether the authors used Asian or WHO cut-off. About ~17 % of the study population is underweight that has differential impact on diabetes. So, I suggest not to combine underweight and normal weight.

Authors’ Response: We agree that some literature has used different BMI categories, particularly for studies in Asia. We, however, decided to use the most widely used and internationally accepted categories of BMI for our analysis. In addition, the Government of Bangladesh also uses these BMI cut-offs for program monitoring and strategic planning purposes. 

We re-analyzed data without combining underweight and normal BMIs and updated the manuscript accordingly.

6. It is not appropriate to combine wealth quintiles.

Authors’ Response: We respectfully disagree with the reviewer’s comment. DHS asset index is a relative index for comparing households or individuals within one survey. Combining wealth quintiles or creating quantiles/terciles from the asset index to differentiate the poor and non-poor respondents is a common, standard practice and published many times in reputable journals (including PLOS ONE). Our re-categorization of the quintiles was primarily based on both old and new study findings, which showed that the households in the two lowest wealth quintiles can be considered as the ‘low socioeconomic status’ for a lower-middle-income country like Bangladesh.

7. It is unclear whether the authors checked multi-collinearity, interaction or effect modification in the data.

Authors’ Response: We checked for multicollinearity, interactions between SES and the covariates, and effect modifications in the dataset. We added texts describing this in pp.9–10, lines 203–208. 

8. When the overall prevalence of diabetes is >10% (especially for table 5). For such a common outcome, logistic regression (odds ratio) overestimates or jeopardizes the association. So, the adjusted association (Odds ratio) might be just by chance, not the true association. Prevalence ratio is the suitable option for a cross-sectional study with a common outcome, while the odds ratio is the best option for a case-control study or the rare outcome. Authors can consult with log-binomial or other advanced methods for calculating adjusted prevalence ratio.

Authors’ Response: Following the Editor’s recommendation, we dropped the section on “Changes in correlates of diabetes between 2011 and 2017-18 among older adults.” The overall prevalence of diabetes in our main analysis remains under 10%, and therefore, we reported the outcomes using logistic regression models.

9. Please provide flowchart of analytic sample. The final report shows that total sample was 12099 (table 13.5.1: women 6918, table13.5.4:men 5181), including pregnant women. However, your analytic sample is 12092 which is only 7 women less compared to the national report.

Authors’ Response: We provided a flowchart in the supplement (Fig S1).

10. For subgroup analysis of women, number of pregnancies/parity should be adjusted in the model.

Authors’ Response: Following the Editor’s recommendation, we dropped the section on “Correlates of diabetes among women of reproductive age.”

11. For table 2, please report overall numerator and denominator.

Authors’ Response: Table 2 already has the denominator (number of respondents) reported. We tried putting the frequencies in each cell, but then the table becomes hard to read. No changes were made.

12. The discussion is sloppy. Policy suggestions for diabetes management need to be compact as currently reported as literature review.

Authors’ Response: We made some adjustments to streamline the discussion section. We would be benefitted from specific comments on how the discussion section can be improved instead of a sweeping statement from the reviewer. 

13. The manuscript must be edited by a native speaker.

Authors’ Response: The manuscript has a native English-speaking and at least one co-author highly proficient in English, and they thoroughly reviewed the manuscript. We request the reviewer to point out the exact places in the revised manuscript where it is felt that further correction is needed to meet the desired standards. 

Comments from Reviewer 2

1. General comments would be to talk about people with diabetes rather than diabetics. I understand from clinical colleagues that there is a move away from labelling people based on their disease status.

Authors’ Response: We thank the reviewer for highlighting this important issue. We have replaced ‘diabetics/diabetic patients’ with ‘people with diabetes’ throughout the manuscript.

2. The sample size is small for doing multiple comparisons, and this should be acknowledged.

Authors’ Response: Following the Editor’s recommendation, we dropped the section on “Changes in correlates of diabetes between 2011 and 2017-18 among older adults.” The analysis in the revised manuscript is based on a representative sample of the adult population in Bangladesh for 2017-18.

3. Please give exact p-values unless p<0.0001.

Authors’ Response: Table 3 has been revised following the PLOS One’s submission guideline. We reported exact p-values for all values greater than or equal to 0.001 and p-values less than 0.001 were expressed as p<0.001. 

4. There are probably >100 p-values in this paper and so it is inappropriate to focus on any <0.05.

Authors’ Response: After dropping the additional analyses as per the Editor’s recommendation, the number of p-values has declined substantially. No changes were made.

5. Have the authors considered doing a global test for heterogeneity in their tables?

Authors’ Response: Testing for heterogeneity is important for meta-analysis or panel data analysis. It helps to deal with different study designs, different interventions, etc., and to take necessary analytical actions. However, following the Editor’s recommendation, our revised manuscript relies on one nationally repetitive cross-sectional survey data. We, therefore, respectfully disagree with the reviewer that performing a test for heterogeneity is essentially applicable to our analysis. 

6. On a similar note, it would be more appropriate to calculate tests for trend for factors that show a trend (eg SES, education, etc).

Authors’ Response: Following the Editor’s recommendation, we dropped the section on “Changes in correlates of diabetes between 2011 and 2017-18 among older adults.” The revised manuscript no longer compares the trends over two time points.

7. Please describe ORs consistently. On line 278, the OR is described as being 31% higher and 2.4 times higher. We do not normally refer to ORs in terms of percentages. Similarly, on line 258 they are referred to as likelihood.

Authors’ Response: We updated the texts (in pp.15–17) in line with the reviewer’s comment.

8. In the Introduction, line 107, please talk about a decline in the probability of dying prematurely (since the probability of dying=1 in all countries).

Authors’ Response: We made the suggested change (p.5, line 107).

9. Line 193: please clarify that ‘hypertensive’ refers to measured hypertensive and so there might be some participants with a diagnosis of hypertension who are controlled by medication. This is fine, but it should be clarified.

Authors’ Response: We updated the texts (in p.9, lines 199–201) to reflect the reviewer’s comment.

10. Table 3: BMI is not nutritional status, please re-label.

Authors’ Response: BMI is a commonly used measure of adult nutritional status for a long time. No changes were made.

11. I would prefer to see SES defined as low, middle, high rather than poor, middle and rich.

Authors’ Response: We made the suggested change.

12. The tables require some work. The use of * to indicate p<0.05, <0.001, etc is inappropriate (as already mentioned). There should be tests for trend and it would be better to use floating absolute risks (Easton et al, 1991, Plummer et al 2004) so that appropriate uncertainty is allocated to the reference group and therefore comparisons can be made between any 2 groups and not just with the reference group.

Authors’ Response: The revised manuscript does not include the 2011 BDHS data. So, the ‘tests for trend’ is no longer an issue. 

In proportional hazards model and conditional logistic regression model for matched case-control studies, floating absolute risk (FAR) is used for estimating the CI for relative risk for each category of a factor variable including the reference group. However, FAR method has several criticisms and we do not find it necessary to achieve the objectives of this study. Therefore, we respectfully disagree with the reviewer and followed the conventional approach (using a reference category) for factor variable.

13. The figures are inadequate. Figure 1 needs N and Cis so that the reader can determine how reliably each prevalence has been determined.

Authors’ Response: We updated Figure 1 by adding CI (Fig-1_R1.svg).

14. Figure 2: the y-axis should be ‘% of those with diabetes’. The current tile implies that 100% of the population aged 18 years or older has diabetes.

Authors’ Response: We updated Figure 2 after incorporating the reviewer’s comment (Fig-2_R1.svg).

---

## [Decision Letter · Decision Letter 1]

18 Jul 2022

PONE-D-21-02392R1

Socioeconomic disparities in diabetes prevalence and management among the adult population in Bangladesh

PLOS ONE

Dear Dr. Ahsan,

Thank you for submitting your manuscript to PLOS ONE. After careful consideration, we feel that it has merit but does not fully meet PLOS ONE’s publication criteria as it currently stands. Therefore, we invite you to submit a revised version of the manuscript that addresses the points raised during the review process.

We look forward to receiving your revised manuscript.

Kind regards,

Mohammad Bellal Hossain

Academic Editor

PLOS ONE

Reviewers' comments:

Reviewer's Responses to Questions

**Comments to the Author**

1. If the authors have adequately addressed your comments raised in a previous round of review and you feel that this manuscript is now acceptable for publication, you may indicate that here to bypass the “Comments to the Author” section, enter your conflict of interest statement in the “Confidential to Editor” section, and submit your "Accept" recommendation.

Reviewer #2: (No Response)

Reviewer #3: All comments have been addressed

2. Is the manuscript technically sound, and do the data support the conclusions?

Reviewer #2: Partly

Reviewer #3: Yes

3. Has the statistical analysis been performed appropriately and rigorously? 

Reviewer #2: Yes

Reviewer #3: Yes

4. Have the authors made all data underlying the findings in their manuscript fully available?

Reviewer #2: Yes

Reviewer #3: No

5. Is the manuscript presented in an intelligible fashion and written in standard English?

Reviewer #2: Yes

Reviewer #3: Yes

6. Review Comments to the Author

Reviewer #2: My major concern remaining is the multitude of p-values in this paper, with no acknowledgement of or correction for multiple testing.

Reviewer #3: The are several details of the statistical analysis missing in the current version of the paper.

1. Table 2 does not include information about the sample size used for the test.

2. The Chi-square test used for table 2 is not sufficiently explained. Please add more details what has been compared and why are there no alternative tests.

3. Table 3 does not include information about the sample size used for the test.

4. Add the mathematical definition of aOR (adjusted odds ratio) used in table 3 to the main text. Explain in detail what has been adjusted.

7. PLOS authors have the option to publish the peer review history of their article (what does this mean?). If published, this will include your full peer review and any attached files.

Reviewer #2: No

Reviewer #3: No

---

## [Author Response · Author response to Decision Letter 1]

1 Sep 2022

Comment from Reviewer 2

1. My major concern remaining is the multitude of p-values in this paper, with no acknowledgement of or correction for multiple testing.

Authors’ Response: For our bivariate analyses (i.e., chi-squared tests), we used the Bonferroni-corrected p-value to examine the association between selected explanatory variables and diabetes prevalence, awareness, treatment, and control (p.13, lines 264–267). Since the multivariable regression analyses in this study were conducted separately for diabetes prevalence, awareness, treatment, and control, we did not correct for multiple testing, and we acknowledged this limitation on p.28, lines 580–583.

Comments from Reviewer 3

1. Table 2 does not include information about the sample size used for the test.

Authors’ Response: Sample sizes (i.e., the number of respondents) are provided in the last row of Table 2 (p.14). 

2. The Chi-square test used for table 2 is not sufficiently explained. Please add more details what has been compared and why are there no alternative tests.

Authors’ Response: We thank the reviewer for this constructive comment. We added texts (and references) on pp.9–10, lines 204–218, to explain our choice of chi-squared tests. 

3. Table 3 does not include information about the sample size used for the test.

Authors’ Response: Sample sizes (i.e., the number of respondents) are provided in the last row of Table 3 (p.17). 

4. Add the mathematical definition of aOR (adjusted odds ratio) used in table 3 to the main text. Explain in detail what has been adjusted.

Authors’ Response: We added texts explaining aORs on p.15, lines 307–314.

---

## [Decision Letter · Decision Letter 2]

14 Oct 2022

PONE-D-21-02392R2Socioeconomic disparities in diabetes prevalence and management among the adult population in BangladeshPLOS ONE

Dear Dr. Ahsan,

Thank you for submitting your manuscript to PLOS ONE. After careful consideration, we feel that it has merit but does not fully meet PLOS ONE’s publication criteria as it currently stands. Therefore, we invite you to submit a revised version of the manuscript that addresses the points raised during the review process.

**Please note that in your manuscript you have stated (Lines 184 to 187) “Body mass index (BMI) was categorized into three groups—underweight/normal if BMI is less than 25 kg/m2, overweight or obese if BMI is 25 kg/m2 or above, and unknown for the respondents whose height and/or weight data were not available. However, in all of your tables, you have presented it as Nutritional status with the following categories: Underweight, Normal, Overweight/obese, and Unknown. I recommend to use to use BMI; not Nutritional status with the following categories: underweight, normal, overweight or obese, and unknown. It will not be good to merge the underweight with normal.  **

We look forward to receiving your revised manuscript.

Kind regards,

Mohammad Bellal Hossain

Academic Editor

PLOS ONE

Journal Requirements:

Reviewers' comments:

Reviewer's Responses to Questions

**Comments to the Author**

1. If the authors have adequately addressed your comments raised in a previous round of review and you feel that this manuscript is now acceptable for publication, you may indicate that here to bypass the “Comments to the Author” section, enter your conflict of interest statement in the “Confidential to Editor” section, and submit your "Accept" recommendation.

Reviewer #2: (No Response)

2. Is the manuscript technically sound, and do the data support the conclusions?

Reviewer #2: Yes

3. Has the statistical analysis been performed appropriately and rigorously? 

Reviewer #2: Yes

4. Have the authors made all data underlying the findings in their manuscript fully available?

Reviewer #2: Yes

5. Is the manuscript presented in an intelligible fashion and written in standard English?

Reviewer #2: Yes

6. Review Comments to the Author

Reviewer #2: I still think the authors have not made clear in table 1 that 'Nutritional status' is based on BMI and vice versa in the text that "Body-mass index (BMI) was categorized into three groups..." is referred to as 'Nutritional status' in Table 1.

7. PLOS authors have the option to publish the peer review history of their article (what does this mean?). If published, this will include your full peer review and any attached files.

Reviewer #2: No

---

## [Author Response · Author response to Decision Letter 2]

27 Nov 2022

Comment from the Editor

1. Please note that in your manuscript you have stated (Lines 184 to 187) “Body mass index (BMI) was categorized into three groups—underweight/normal if BMI is less than 25 kg/m2, overweight or obese if BMI is 25 kg/m2 or above, and unknown for the respondents whose height and/or weight data were not available. However, in all of your tables, you have presented it as Nutritional status with the following categories: Underweight, Normal, Overweight/obese, and Unknown. I recommend to use to use BMI; not Nutritional status with the following categories: underweight, normal, overweight or obese, and unknown. It will not be good to merge the underweight with normal.

Authors’ Response: We are sorry for this oversight. Based on the reviewer’s comments in an earlier review, we re-analyzed the data using four categories of BMI, and underweight with normal are no longer merged in the current analyses. We have corrected the variable description on p.8, lines 181–182. We also replaced ‘Nutritional status’ with ‘BMI’ throughout the manuscript. 

Comment from Reviewer 2

1. I still think the authors have not made clear in table 1 that 'Nutritional status' is based on BMI and vice versa in the text that "Body-mass index (BMI) was categorized into three groups..." is referred to as 'Nutritional status' in Table 1.

Authors’ Response: We have corrected the variable description on p.8, lines 181–182. We also replaced ‘Nutritional status’ with ‘BMI’ throughout the manuscript.

---

## [Editor Report · Decision Letter 3]

5 Dec 2022

Socioeconomic disparities in diabetes prevalence and management among the adult population in Bangladesh

PONE-D-21-02392R3

Dear Dr. Ahsan,

We’re pleased to inform you that your manuscript has been judged scientifically suitable for publication and will be formally accepted for publication once it meets all outstanding technical requirements.

Kind regards,

Mohammad Bellal Hossain

Academic Editor

PLOS ONE
---

## [Editor Report · Acceptance letter]

12 Dec 2022

PONE-D-21-02392R3 

Socioeconomic disparities in diabetes prevalence and management among the adult population in Bangladesh 

Dear Dr. Ahsan:

I'm pleased to inform you that your manuscript has been deemed suitable for publication in PLOS ONE. Congratulations! Your manuscript is now with our production department. 

Kind regards, 

on behalf of

Dr. Mohammad Bellal Hossain 

Academic Editor

PLOS ONE